# Analyzing Pixel-Level Relationships between Luojia 1-01 Nighttime Light and Urban Surface Features by Separating the Pixel Blooming Effect

**Ji Wu [1], Zhi Zhang [2], Xiao Yang [2] and Xi Li [1,*]**

[1]  State Key Laboratory of Information Engineering in Surveying, Mapping and Remote Sensing, Wuhan University, Wuhan 430079, China; wuji1996@whu.edu.cn

[2]  Institute of Geophysics & Geomatics, China University of Geosciences, Wuhan 430074, China; 3slab@cug.edu.cn (Z.Z.); 1201820419@cug.edu.cn (X.Y.)

*  Correspondence: lixi@whu.edu.cn; Tel.: +86-180-6241-1350

**Abstract:** Nighttime light (NTL) remote sensing data can effectively reveal human activities in urban development. It has received extensive attention in recent years, owing to its advantages in monitoring urban socio-economic activities. Due to the coarse spatial resolution and blooming effect, few studies can explain the factors influencing NTL variations at a fine scale. This study explores the relationships between Luojia 1-01 NTL intensity and urban surface features at the pixel level. The Spatial Durbin model is used to measure the contributions of different urban surface features (represented by Points-of-interest (POIs), roads, water body and vegetation) to NTL intensity. The contributions of different urban surface features to NTL intensity and the Pixel Blooming Effect (PIBE) are effectively separated by direct effect and indirect effect (pseudo-$R^2$ = 0.915; Pearson correlation = 0.774; Moran's I = 0.014). The results show that the contributions of different urban surface features to NTL intensity and PIBE are significantly different. Roads and transportation facilities are major contributors to NTL intensity and PIBE. The contribution of commercial area is much lower than that of roads in terms of PIBE. The inhibitory effect of water body is weaker than that of vegetation in terms of NTL intensity and PIBE. For each urban surface feature, the direct contribution to NTL intensity is far less than the indirect contribution (PIBE of total neighbors), but greater than the marginal indirect effect (PIBE of each neighbor). The method proposed in this study is expected to provide a reference for explaining the composition and blooming effect of NTL, as well as the application of NTL data in the urban interior.

**Keywords:** Luojia 1-01; nighttime light; urban surface; points-of-interest; pixel blooming effect; spatial autoregressive model

## 1. Introduction

Cities are complex dynamic systems composed of infrastructure, human activities and social connections [1]. More than half of the world's population now live in cities, which also brings increasingly complex urban problems [2]. As an effective measure of urban monitoring, remote sensing includes daytime and nighttime remote sensing. Daytime remote sensing primarily reveals the information of natural environment and artificial surface. By contrast, nighttime remote sensing reveals human socio-economic activities [3].

Nighttime light (NTL) remote sensing can effectively reveal the human activities in urban development. NTL remote sensing data can record the light from buildings, roads and vehicles, etc., at night by satellite sensors [4]. It has a statistically significant correlation with human activities such as population distribution, economic growth and infrastructure [5]. Therefore, it is widely used in urbanization monitoring to estimate GDP, population, electrical consumption, the extraction of built-up areas and the estimation of housing vacancy rate [6–8]. However, due to the coarse spatial resolution and blooming

effect of NTL data, current studies are more often applied to the outside of the city. In order to provide a reference for the NTL application in the urban interior, it is crucial to comprehensively understand the composition of NTL and the factors influencing NTL variations [9]. Currently, commonly used NTL data are observed by various sensors and platforms, such as the Defense Meteorological Satellite Program-Operational Linescan System (DMSP-OLS) [10] and Suomi National Polar-Orbiting Partnership Satellite's Visible Infrared Imaging Radiometer Suite (NPP-VIIRS) [11]. As a new generation of NTL satellite, Luojia 1-01 was launched by Wuhan University of China on 2 June 2018 [12]. Compared with DMSP-OLS and NPP-VIIRS, Luojia 1-01 NTL data provide more abundant details [13]. It provides a solution for the application of NTL data in the urban interior.

The relationships between NTL and different land use/land cover (LULC) types have been explored in previous studies [9,14–16], which is essential for exploring the source, composition and application of NTL in the urban interior. Li et al. developed an unmixing model to quantify the land use (produced by LULC) contribution to NTL [14]. Ma focused on the relationships between urban surfaces (produced by LULC and Point-of-interest (POI)) and NPP-VIIRS NTL intensity at the pixel level [15]. However, due to the coarse spatial resolution, DMSP-OLS and NPP-VIIRS data may not be able to distinguish different LULC especially in the urban interior. Wang et al. explored the relationships between artificial surfaces and NTL intensity at the parcel level. It is concluded that NTL variations in Luojia 1-01 data for different artificial surfaces (produced by LULC and POIs) were more significant than those in NPP-VIIRS data [9]. In brief, the features representing the characteristics or background environment of the urban surface illuminants affect the NTL variations to a greater or lesser extent, whether they are LULC, POIs or urban surface [9,15]. Nevertheless, due to the smoothness of the NTL details caused by blooming effect, the pixel-level research of NTL in the urban interior remains rare.

The blooming effect means that the illuminated area observed by the sensor is wider than the geographic area of the illuminant, which leads to a wider extent of illuminated area displayed by the image than the actual extent [17,18]. The blooming effect has been observed in the current NTL images, such as DMSP-OLS, NPP-VIIRS and Luojia 1-01 data [9,19,20]. Light from human settlements spread is far beyond the extent of illuminants. The light of coastal cities can be observed 20 km away from the coastline. For cities with clear boundaries, the illuminated area usually expands the urban size by 10 times [17]. The reason for these phenomena is precisely the blooming effect. For the blooming effect, there are various causes and factors, as follows. (i) Field-of-view variation: each elliptical ground scanning area captures the light from multiple neighboring illuminants, which results in a significant overlap between neighboring pixels [21]. (ii) Geolocation error [21]. (iii) Atmospheric scattering: the sky is illuminated by the overflow of artificial illuminant from the land surface [22]. Atmospheric scattering is one of the reasons affecting the blooming effect of NTL images [17,21]. (iv) Accumulation of (i)–(iii) in the annual/monthly NTL data [21]. (v) Characteristics of the land surface illuminants: the dimensions [19] and brightness intensity [21,23] of land surface illuminants affect the blooming effect's intensity of NTL image. (vi) Background environment of the land surface illuminants: vegetation has an inhibitory effect on NTL [24]. By contrast, the blooming effect is more significant around water bodies and snow surfaces [25]. The blooming effect expands the extent of the illuminated area and reduces the simulation accuracy of the population distribution and built-up area extraction [9]. The blooming effect has given rise to a succession of adverse impacts on NTL-based applications, especially in city centers, where the detailed information is needed.

In previous studies, the blooming effect was assumed as the impact between the built-up area and the non-built-up area. However, while the study area is located in the urban interior, it is necessary to divide the blooming effect into more detailed components of impact between various urban surface components. Due to the blooming effect, the detected light of a center pixel appears to "spill over" to its neighboring pixels, while lights from its neighboring pixels also scatter back to the center pixel [26]. Zheng et al. eliminated

the pixel blooming effect of the built-up area and non-built-up area in DMSP-OLS data through the mutual influence of lights from a pixel and its neighbors [26]. However, this hypothesis is not enough to measure the blooming effect of NTL data in the urban interior. The method in Abrahams et al. is able to distinguish built-up areas and non-built-up areas around the urban boundary, but it is not able to distinguish them in the urban interior [21]. Zheng et al. assumed that all pixels were subject to the same kind of pixel blooming effect, but did not consider the differences in blooming effect caused by the characteristics of the land surface illuminants [26]. Recent studies suggest that there are obvious differences in NTL intensity in the regions of different LULC [9], and the characteristics of the land surface illuminants and the background environment of the land surface illuminants affect the blooming effect to a certain extent [19,21,23,24]. The blooming effect was omitted, in the studies of analyzing the relationships between NTL intensity and urban surface features. In previous studies related to simulating and eliminating the blooming effect, the blooming effect of different urban surfaces is usually not distinguished. Because of these reasons, most investigations are applied to regional-level or sub-regional-level surveys of human activity. Therefore, while the urban surface is divided into more detailed components (such as commercial, road, water body or vegetation area), the NTL intensity and the blooming effect generated by different urban surfaces may also be different. For different urban surfaces, how much of the NTL is contributed by the central pixel's urban surface, and how much of the NTL is contributed by the blooming effect of neighboring pixels? This is one of the problems to be solved in this paper.

The purposes of this study are to: (i) Propose a method through which to measure the direct and indirect contribution of various urban surface features to the Luojia 1-01 NTL intensity. (ii) Quantitatively measure the intensity of the Pixel Blooming Effect (PIBE) generated by different urban surface features. The article is organized as follows. Section 2 describes the experiment area and the related datasets. Section 3 provides an overview of the research procedures, including the definition of PIBE, data preprocessing, the spatial autoregressive model and the measurement of spatial autocorrelation. In Section 4, the test of different neighboring effects, the Spatial Durbin model (SDM) fitting, and the spatial partitioning of feature contributions to NTL intensity or PIBE are displayed. Section 5 discusses the contribution of various urban surface features and the limitations of our method. Finally, Section 6 summarizes the significance and future prospects of this article.

## 2. Study Area and Datasets

### 2.1. Study Area

The study area of this study is the urban area within the Third Ring Road of Wuhan (Figure 1). Wuhan, which is the capital of Hubei Province in central China, is located at 113°41′ E~115°05′ E, 29°58′ N~31°22′ N. The left of Figure 1 illustrates the administrative boundary and the Luojia 1-01 NTL image of Wuhan. In Wuhan, water area covers around 26.1% of the municipal administrative area, and the dense buildings and NTL cover the ground in the urban interior. Due to the diversified ground objects, it is a typical area in which to investigate NTL and PIBE. The urban areas within the Third Ring Road were selected to explore the relationships between urban surface features and NTL at the pixel-level.

### 2.2. NTL Data

The NTL data applied in this study are the Luojia 1-01 data of Wuhan on 13 June 2018, and its local overpass time is 22:41. Currently, various studies that revolve around the quality and applications of Luojia 1-01 NTL data are underway with new harvests, which demonstrates the advantages of these data [27–29], such as increased potential in urban monitoring [30]. Compared with DMSP-OLS data and VIIRS-DNB data, Luojia 1-01 provides NTL data with higher quantization and finer spatial details (Table 1) [31]. Therefore, it provides convenience for exploring more detailed NTL composition and structure in the urban interior. In Section 3.2, we report on our performance of geometric

correction and radiometric calibration on a Luojia1-01 NTL image. After standardization, NTL intensity is obtained as the value of dependent variable. Every image and all the spatial data applied in this study were projected to "WGS_1984_UTM".

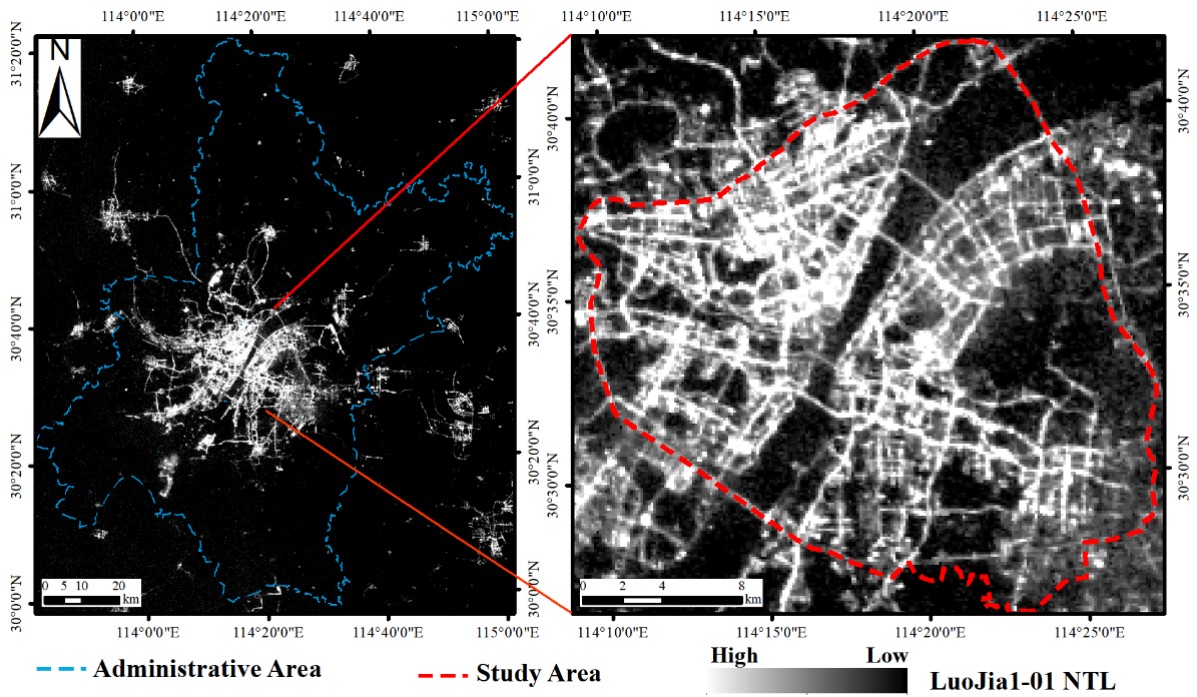

**Figure 1.** NTL image of Luojia 1-01 in the study area.

**Table 1.** Comparison of different NTL data.

| Sensor | DMSP/OLS | VIIRS/DNB | Luojia 1-01 |
| --- | --- | --- | --- |
| Temporal resolution | Global coverage can be obtained every 24 h | Daily images can be downloaded | 15 day revisit time |
| Spectral band | 500–900 nm | 500–900 nm | 460–980 nm |
| Quantization | 6 bits | 14 bits | 14 bits |
| Spatial resolution | 3000 m | 740 m | 130 m |
| Time span | 1992–2013 | 2011–present | 2018–2019 |

### 2.3. Urban Surface Feature Datasets

The urban surface features defined in this study are composed by overlying 13 maps. A total of 11 maps of artificial facilities represent the characteristics of the urban surface illuminants, including POIs (6 class, Figure 2c) and roads (5 level, Figure 2d). Two maps of natural surfaces represent the background environment of the urban surface illuminants, including the water area and vegetation (Figure 2b). Figure 2 illustrates the spatial distribution of water area, vegetation, POIs and roads in the study area.

#### 2.3.1. Natural Surface from Landsat 8

In this study, the background environment of the urban surface illuminants is divided into three major categories (water body, vegetation and artificial surface). Due to the dummy variable trap, only two dummy variables (i.e., water body and vegetation) were applied into the model. Landsat-8 multispectral images were used to extract water and vegetation in this study (http://www.gscloud.cn/, accessed on 18 December 2018). After radiometric correction and atmospheric correction, the normalized difference water index (NDWI) and normalized vegetation index (NDVI) were used to extract the water and vegetation. The results were resampled to 130 m × 130 m.

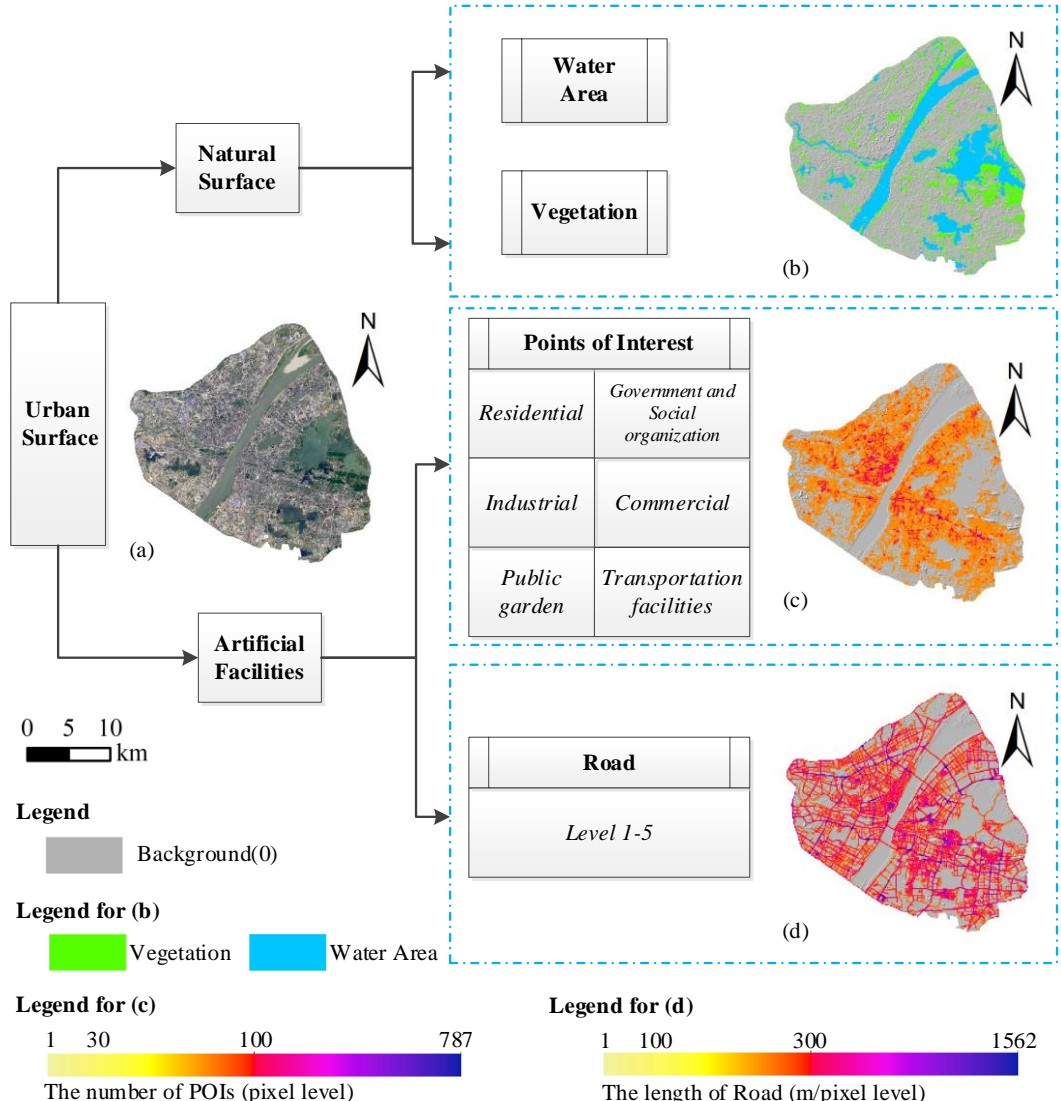

**Figure 2.** Spatial distribution of independent variables: (**a**) Landsat 8 image; (**b**) natural surface spatial distribution map: include water area and vegetation; (**c**) POI density map: composed with all types of POIs; (**d**) road density map: composed with all levels of roads.

### 2.3.2. Artificial Facilities from POI

Artificial facilities, such as POIs and roads, were used to characterize the characteristics of the urban surface illuminants. The initial POI data were collected from Amap (https://www.amap.com/, accessed on 25 December 2017), which includes 23 types. In this study, we reclassified it into 6 categories (i.e., residential, government and social organization, industrial, commercial, public garden, transportation facilities) with a total of 301,331 points (Table A1). Although spurious social data may arise, the spatial distribution can be accurately reflected by using a number of points.

### 2.3.3. Artificial Facilities from OpenStreetMap

The initial road network data was collected from OpenStreetMap (OSM, http://www.openstreetmap.org, accessed on 18 December 2018), which includes 22 types, such as motorway, trunk, primary road, and secondary road. In this study, we reclassified them into five categories (i.e., road1, road2, road3, road4, road5), with a total length of 3681.7 km (Table A2). For Wuhan, underground roads (such as tunnels and subways) were removed.

## 3. Methods

This study used Luojia 1-01 NTL and urban surface features (13 variables from water body, vegetation, POIs and Roads) to build a spatial autoregressive model at pixel level. Figure 3 illustrates the processes of this article. Based on the research basis of previous studies on the NTL interaction between different areas, hypothesis testing was used to determine which neighboring effects should be put into the model. Finally, according to the direct and indirect effects, we explored the contributions of different urban surface features to NTL intensity and PIBE.

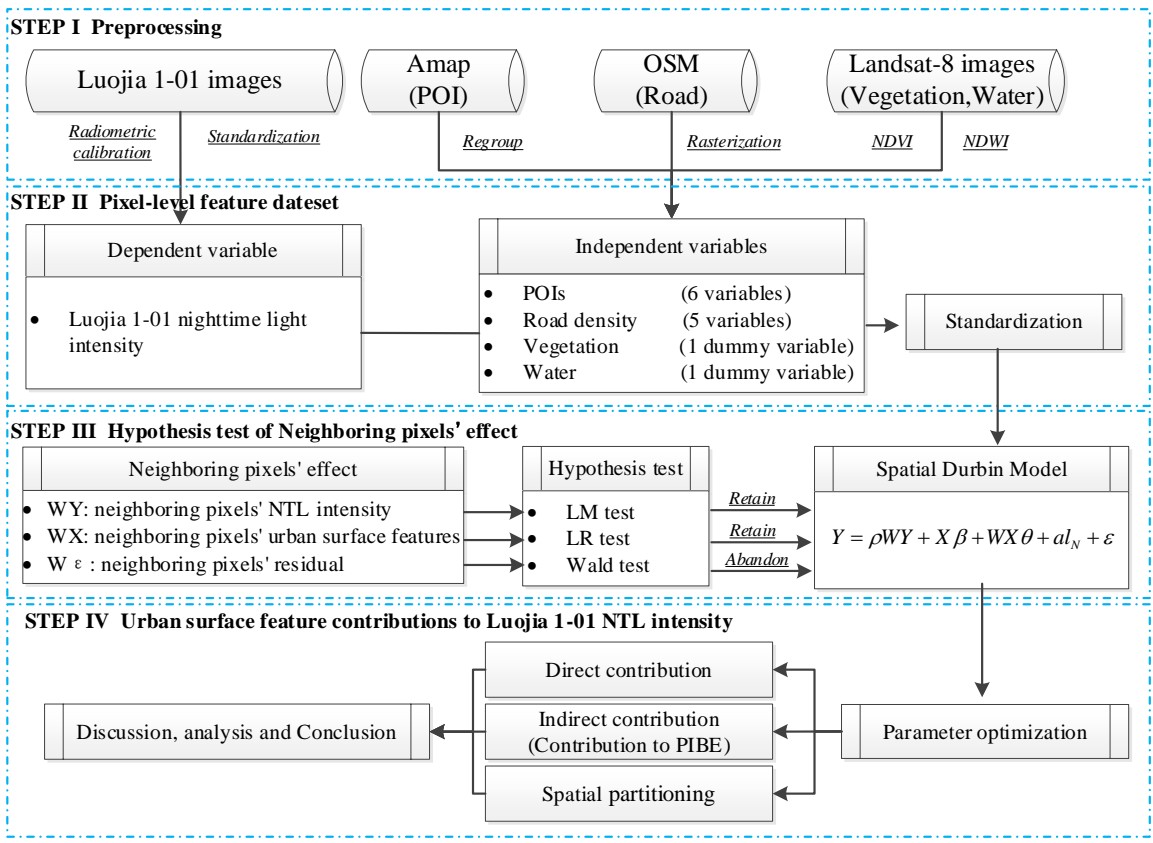

**Figure 3.** Flowchart.

### 3.1. The Pixel Blooming Effect (PIBE)

Only a small number of studies have performed a quantitative discussion on blooming effect [21,26,32]. Thus, we needed a term with which to clarify the blooming effect in this study, which means a few NTL of a central pixel "overflows" from its neighboring pixels. In this study, we continue to use the term, "Pixel Blooming Effect (PIBE)", to explain the blooming effect of the NTL images [26]. The PIBE emphasizes the fact that all pixels feature a blooming effect, not just in suburbs. Due to their differing methodologies, PIBE and traditional blooming effect [17] need to be distinguished. Due to the PIBE, not only does the NTL of the center pixel seem to "spill over" to its neighboring pixels, but the NTL of its neighboring pixels also "spills" back to the center pixel [26]. What this study explores is how much of the central pixel's NTL "spilled" from its neighboring pixels, and how much of the central pixel's NTL is caused by the central pixel's urban surface.

### 3.2. Data Preprocessing

In order to accurately discuss the NTL and the PIBE, the Luojia 1-01 NTL image performed geometric correction and radiometric calibration. The road network was clearly visible, due to the high spatial resolution of Luojia 1-01 data. For each image, we used

road network intersections as control points for geometric correction. For the radiation calibration, Equation (1) based on laboratory calibration provided by the satellite data producer is used to convert the digital number into the radiance.

$$L = 10^{-10} DN^{3/2} \omega \tag{1}$$

where $DN$ denotes the digital numbers of a pixel; $L$ denotes the radiance, with unit $nWcm^{-2}sr^{-1}$; and $\omega$ denotes the bandwidth. The radiometric range of the Luojia 1-01 is 460–980 nm, so that $\omega = 5.2 \times 10^{-7} m$.

For city surface, we obtained an urban surface feature map of 32,104 rectangular fishing nets by preprocessing. The size of each fishing net was 130 m × 130 m, with 13 independent variables (i.e., water, vegetation, residential, government and social organization, industrial, commercial, public garden, transportation facilities, road1, road2, road3, road4, road5). Water and vegetation are set as dummy variables, that is, for all grids, the value was 0 when there is no water area and 1 when there was a water area. For each grid, the density of the urban surface was determined by the count of each class of POI and the length of each level road.

We explored a total of 13 variables, each of which featured different units and ranges. To facilitate the comparison of their respective contributions, we normalized each variable (except water and vegetation) using Equation (2).

$$X_r = \frac{X_{lr0} - \mu_{lr0}}{\sigma_{lr0}}; X_{lr0} = LN(X_{r0} + 1) \tag{2}$$

where $X_r$ is the standardized value for the $r$-th independent variable (representing the value of the $r$-th urban surface); $X_{r0}$ is the initial value of the $r$-th independent variable (representing the density of the $r$-th urban surface); $X_{lr0}$ is the logarithmically transformed value for $X_{r0}$; and $\mu_{lr0}$ is the mean value for $X_{lr0}$; $\sigma_{lr0}$ is the standard deviation for $X_{lr0}$.

### 3.3. Ordinary Least Squares Regression Model

In most spatial analyses, the general method is to start with a non-spatial linear regression model, and then to test whether this model needs to be extended to a model with spatial effects [33]. This kind of non-spatial linear regression model usually uses ordinary least squares (OLS) for estimation, so it is usually called the OLS model. The form is as Equation (3):

$$Y = X\beta + al_N + \varepsilon, \quad \varepsilon \sim N\left(0, \sigma^2 I_N\right), iid \tag{3}$$

where $Y$ is the dependent variable; $X$ is the independent variable; $\beta$ is the coefficient to be estimated; $N$ is the sample size; $a$ is the constant; $l_N$ is the unit vector; and $\varepsilon$ is the random error term, which obeys independently identically distribution. The residual is usually used as the estimated value of the random error term.

The assumption that the observations or residuals are independent of each other greatly simplifies the model. However, in the spatial context, this simplification seems far-fetched [33]. Spatial dependence simply reflects the situation of spatial data, that is, the observed value of a pixel depends on the observed value of its neighboring pixels. This greatly reduces the estimation accuracy of OLS model. Ma used the OLS model to discuss the pixel-level relationship between urban surfaces and NPP-VIIRS NTL intensity, and observed that the goodness of fit between them was only $R^2 = 0.16$ [34]. Therefore, it is usually necessary to measure and test the spatial autocorrelation of the residual (by Moran's I). Next, the Lagrange multiplier test and the likelihood ratio test are used to determine the specification of the spatial autoregressive model so as to expand the OLS model into the spatial autoregressive model.

### 3.4. Spatial Autoregressive Models

How NTL "spills over" to its neighboring pixels is one of the problems to be solved in this study. The representation of PIBE on the image is that the NTL intensity of the central pixel is affected by the urban surface or the NTL intensity of the neighboring pixels. In the spatial autorepression model, we call this effect the spillover effect. The spatial autoregressive model is a model used to deal with the spillover effect or PIBE between geographical units. It is a new tool to solve the problem of how spatial variables overflow to their neighboring units [35]. In the spatial autoregressive model, three kinds of neighboring effects are usually used to solve the spillover effect or PIBE, that is, the neighboring effects between the dependent variable, the independent variable and the residual [36]. Zheng et al. eliminated the blooming effect of DMSP-OLS by neighboring the pixels' NTL intensity [26]. The spatial autoregressive model is also applied to the simulation of light emissions and skyglow. Daniel et al. separated the direct and indirect contributions of various geographical independent variables [34].

Neighboring effects in the spatial autorepression model are divided into three categories, as follows. (i) $WY$: the neighboring effect between the dependent variables of different pixels; (ii) $WX$: the neighboring effect between the independent variables of different pixels; (iii) $W\varepsilon$: the neighboring effect between the residuals of different pixels [36]. In general, three different neighboring effects can explain why the observations in one pixel are dependent on those in another pixel. A complete model with all the types of neighboring effect is called the General Nesting Spatial model (GNS, Equation (4)) [37].

$$Y = \rho WY + X\beta + WX\theta + al_N + \varepsilon, \quad \varepsilon = \lambda W\varepsilon + v \tag{4}$$

where $Y$, $X$ and $al_N$ are consistent with Equation (3); $W$ represents spatial weight matrix; $\rho$ represents the spatial autoregressive coefficient; $\beta$ is the coefficient to be estimated for the $X$; $\theta$ is the coefficient to be estimated for the $WX$; $\lambda$ represents the spatial autocorrelation coefficient; $\varepsilon$ represents the error term; and $v$ represents the trait components, including the components unexplained by spatial dependence.

The spatial autoregressive model includes a series of models. By eliminating different neighboring effects, GNS can be simplified into seven other models, as shown in Figure 4. The OLS model is the most specialized model (i.e., when $\rho$, $\theta$, $\lambda$ are equal to 0). In Section 4.2, we performed hypothesis tests (i.e., the Moran test [38], the Lagrange multiplier (LM) test [35,39], the Wald test and the Likelihood ratio (LR) test [35]) for three neighboring effects to determine which neighboring effect should be included into model. This was in order to determine which model specification was closest to the real generation process of NTL intensity and PIBE.

After the hypothesis testing and model screening described in Section 4.2, this study concludes that the optimal model is the Spatial Durbin model (SDM, Equation (5)) [37]. The spatial autoregressive model expands the information by leading the information (or observations) of neighboring pixels into the model. This means that the observations do not satisfy the independence assumption. Therefore, the regression coefficients (such as $\rho$, $\beta$ and $\theta$) are usually not used to measure the contributions of the independent variable to the dependent variable [37]. In order to observe these contributions, the total effect, the direct effect and the indirect effect (Table S1) are usually employed [37]. For the center pixel $i$ and its neighbor pixel $j$, the total effect includes the direct effect (the variation of independent variable $X_i$ affects the dependent variable $Y_i$) and the indirect effect (the variation of independent variable $X_i$ affects the dependent variable $Y_j$). In this study, the total effect is interpreted as the total contribution of urban surface features ($X$) to NTL intensity ($Y$). The direct effect (or direct contribution) is interpreted as the contribution of the center pixel's urban surface features ($X_i$) to the center pixel's NTL intensity ($Y_i$). The indirect effect (or indirect contribution) is interpreted as the contribution of the center pixel's urban surface features ($X_i$) to the neighboring pixels' NTL intensity ($Y_j$). The direct effect measures the NTL intensity determined by the center pixel's urban surface features, and the indirect effect measures the PIBE. It is worth noting that the effects from an observation and the

effects to an observation are numerically equal, because the spatial weight matrix in this study is a symmetric matrix.

$$Y = \rho WY + X\beta + WX\theta + al_N + \varepsilon \tag{5}$$

where the $Y$, $X$, $W$, $\beta$, $al_N$, and $\varepsilon$ are consistent with Equation (4). Their realistic meaning is as follows: $Y$ represents the NTL intensity; $W$ represents the adjacent relationship of pixels; $WY$ is the average of the neighboring pixels' NTL intensity; $\rho$ is the coefficient of $WY$; $X$ represents the urban surface features (i.e., water, vegetation, residential, government and social organization, industrial, commercial, public garden, transportation facilities, road1, road2, road3, road4, road5); $WX$ is the average of the neighboring pixels' $X$; and $\theta$ is the coefficient of $WX$.

$$W_{ij} = \begin{cases} 1, bound(i) \cap bound(j) \neq \varnothing \\ 0, bound(i) \cap bound(j) = \varnothing \end{cases} \tag{6}$$

where $bound()$ represents the boundary of pixel; $W_{ij}$ represents the adjacency relationship between pixel $i$ and pixel $j$, while $W_{ij} = 1$ indicates that two pixels are neighbor, and $W_{ij} = 0$ indicates that two pixels are not neighbor. In this study, $W$ is an N*N rook spatial weight matrix. The value N represents the sample size.

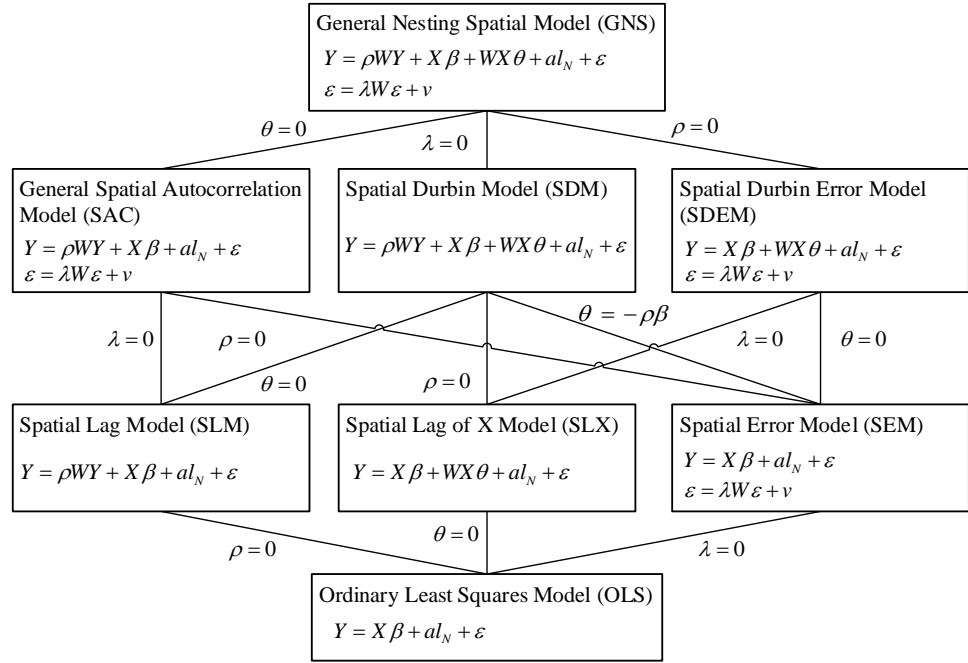

**Figure 4.** Relationship between different spatial autoregressive models. Due to over parameterization, the GNS model and the SAC model are very difficult to explain. Thus, they are rarely applied in empirical research [40]. Therefore, we do not discuss them in this article.

*3.5. Moran's I*

Cliff et al. applied Moran's $I$ to test whether there is a spatial autocorrelation in the regression residual so as to point out the error of model specification caused by spatial dependence [38]. It is necessary to determine whether the PIBE in the dependent variable is completely explained by the independent variable, that is, whether there is unexplained PIBE in the residual. Therefore, the residual is usually tested by Moran's $I$ (Equations (7) and (8)).

$$I = \frac{N}{\sum_{i=1}^{N} \sum_{j=1}^{N} W(i,j)} \frac{\sum_{i=1}^{N} \sum_{j=1}^{N} W(i,j)(X_i - \overline{X})(X_j - \overline{X})}{\sum_{i=1}^{N} (X_i - \overline{X})^2} \tag{7}$$

$$Z(I) = (I - E(I)) / \sqrt{var(I)} \qquad (8)$$

where $I$ is the spatial autocorrelation index; $N$ is the sample size; $X_i$ is the observed value of the sample $i$; $\overline{X}$ is the average of $X$. $W(i, j)$ is the value of the spatial weight matrix between $i$ and $j$; $E(I)$ is the expectation of $I$; and $var(I)$ is the variance deviation of $I$. If $|Z(I)|$ is greater than 1.65 and the $p$-value is less than 0.05, this indicates that the $I$ is significant.

In this study, the regression residuals of each model are used to calculate the Moran's I. The range of $I$ is between [–1, 1]. If $I < 0$, it means spatial discretization. If $I > 0$, it means spatial clusters. If $I = 0$, it means that there is no spatial correlation or spatial random distribution.

## 4. Results

### 4.1. Exploratory Statistical Analysis

Without considering neighboring pixels, common statistics can display an intuitive comparison of the NTL radiance of different urban surfaces. Figure 5a compares the NTL radiance of different urban surfaces. According to Figure 5a, the pixels where roads and transportation facilities are located produce significantly more light than other pixels (the median is greater than 100). Considerable natural surfaces (such as water and vegetation) are usually illuminated at night, with the median NTL radiance clearly greater than 0. This phenomenon is usually due to the PIBE of NTL.

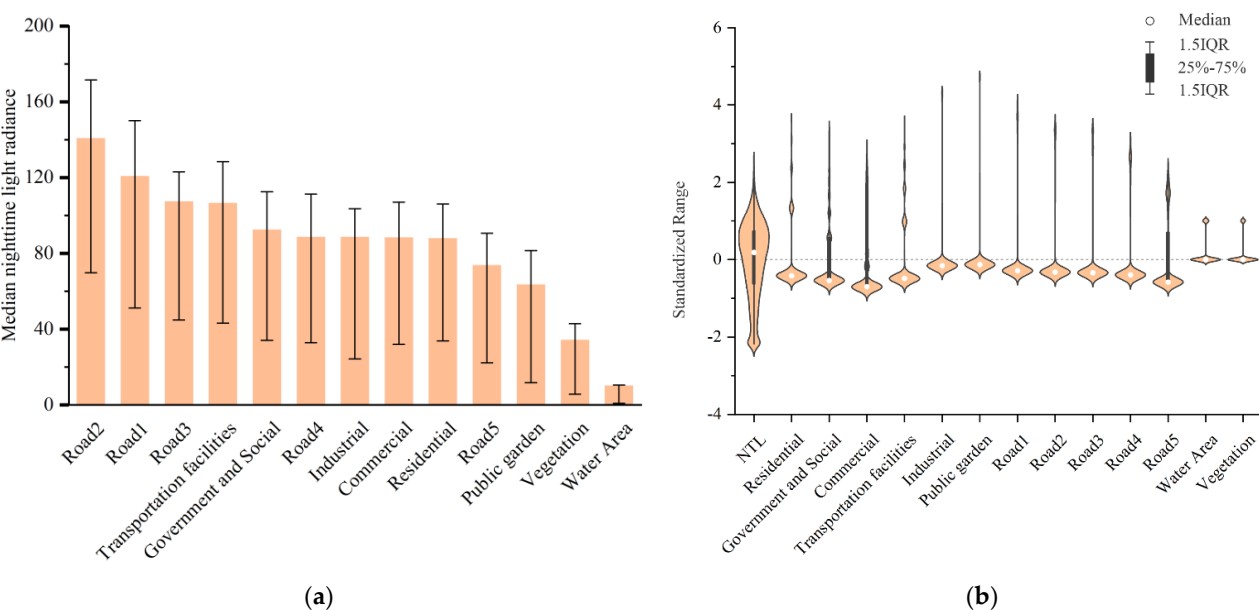

**Figure 5.** Exploratory statistical analysis: (**a**) NTL radiance of different urban surface in $nWcm^{-2}sr^{-1}$ (here, the lower quartile, median, and upper quartile are presented); (**b**) the violin plot of different urban surface feature.

In our urban surface feature map, there are a total of 32,104 samples. After the logarithmic transformation and normalization of $Y$ and $X$, we produced a violin plot (Figure 5b). According to Figure 5b, for all the artificial facilities, most of the independent values were below the mean value, which equals 0. Because there are always a few artificial facilities densely distributed in a few pixels in the urban interior, each independent variable features a small number of high outliers. For the natural surface, we used dummy variables whose value was 0 or 1. For the NTL radiance, its value was mainly within $\pm 3\sigma$.

### 4.2. Spatial Autoregressive Model Hypothesis Test

It is usually necessary to determine which kind of neighboring pixel effect should be included in the model by using hypothesis tests. This determination is carried out in order to determine which model specification is closest to the real generation process of NTL

intensity and PIBE. This section is summarized as follows. Section 4.2.1 applies the Moran test to the regression residuals and concludes that the model needs to consider the effect of neighbor pixels. In Section 4.2.2, it is statistically determined by the hypothesis test (i.e., LM test, Wald test and LR test) that the NTL of neighbor pixels (*WY*) should be included in the model. In Section 4.2.3, it is statistically determined by the hypothesis test (i.e., the LR test) that the urban surface of neighbor pixels (*WX*) should be included in the model. It is concluded that the optimal model is SDM. This process is expanded in this section.

### 4.2.1. Hypothesis Test of Neighboring Pixels' Effect

In this study, Moran's I was used to test whether the model needed to consider the effect of neighboring pixels [38]. Firstly, Moran's I was applied to the OLS model and SLX model regression residual. According to Table 2, the adjusted-$R^2$ of the OLS model was 0.45, indicating that the urban surface features explained 45% of the NTL intensity variation. The Moran's I of the OLS regression residual was 0.685 (*p*-value < 0.01), indicating that the null hypothesis of "no spatial autocorrelation" was rejected at the significance level of 1%, that is, there was significant spatial autocorrelation in the residual. Similarly, Table 2 illustrates that the adjusted-$R^2$ of the SLX model was 0.56, which was higher than the adjusted-$R^2$ of the OLS model (0.45), indicating that adding the neighboring pixels' urban surface (*WX*) to the model may be a statistically better solution. However, the Moran's I of the SLX regression residual was 0.732 (*p*-value < 0.01), indicating that there was a significant spatial autocorrelation in the SLX regression residual. Therefore, neighbor pixels need to be considered to deal with the spatial autocorrelation problem of model residuals. Therefore, it is necessary to consider the neighboring pixels' effect.

**Table 2.** Moran's I and LM test of OLS and SLX model regression residuals.

| Model | Moran's I | Ajusted-$R^2$ | LM-Lag | Robust LM-Lag | LM-Error | Robust LM-Error |
|-------|-----------|---------------|--------|---------------|----------|-----------------|
| OLS | 0.685 ** | 0.45 | 35,769.6 ** | 6056.5 ** | 29,760.4 ** | 47.3 ** |
| SLX | 0.732 ** | 0.56 | 35,485.9 ** | 1812.1 ** | 34,049.1 ** | 375.3 ** |

** represents *p*-value < 0.01.

### 4.2.2. Hypothesis Test of Neighboring Pixels' NTL

The Moran test demonstrated that the effect of neighboring pixels should be considered, but this test cannot determine whether the NTL intensity of neighbor pixels (*WY*) or the error term of neighbor pixels (*Wε*) should be included in the model. According to a previous study [35], it needs to be judged by using the LM test (Table 2). Both LM-Lag and LM-Error are significant at 1%. In this situation, it needs to be further judged by the robust LM-Lag and robust LM-Error [39]. However, the robust LM-Lag and the robust LM-Error are also significant at 1%. In this case, the model with larger robust LM value was usually better in the previous study, which is robust LM-Lag. For the OLS model and the SLX model, the model specification with *WY* (i.e., SLM and SDM) is better than the model specification with *Wε* (i.e., SEM and SEDM).

After confirming that the model with the neighboring pixels' NTL intensity (*WY*) is better, it is necessary to determine whether the neighboring pixels' NTL intensity is significant. Anselin et al. provide three methods based on the maximum likelihood theory, namely the Wald test, the LR test and the LM test [35]. The size arrangement of the three statistics should follow Wald > LR > LM. Otherwise, the model specification is incorrect. Table 3 displays the test results of the SLM model and SDM model. The three statistics in Table 3 completely follow this size arrangement order, with a *p*-value < 0.01. This indicates that the neighboring pixels' NTL intensity (*WY*) in the SLM model and the SDM model was significant, and that the model could not be simplified to the OLS model or the SLX model.

It can be concluded that the model specification with the neighboring pixels' NTL intensity (*WY*) is appropriate, which is closer to the real generation process of NTL.

**Table 3.** Wald, LR and LM tests of SLM and SDM models.

| Model | Wald Test | LR Test | LM Test |
|---|---|---|---|
| SLM | 206,437.6 ** | 49,988.9 ** | 35,769.6 ** |
| SDM | 150,180.3 ** | 43,034.5 ** | 35,485.9 ** |

(1) ** represents *p*-value < 0.01; (2) $H_0 : \rho = 0$; $H_1 : \rho \neq 0$; $\rho$ is the coefficients of $WY$.

### 4.2.3. Hypothesis Test of Neighboring Pixels' Urban Surface Features

The difference between the SDM model and the SLM model is in whether the neighboring pixels' urban surface features ($WX$) should be included in the model. The LR test can effectively test this problem [35], with the null hypothesis $H_0 : \theta = 0$. The statistic obeys $\chi^2$ distribution with degree of freedom $k$ The value $\theta$ is the coefficient of $WX$. The value $k$ is the number of parameters to be estimated. The result is LR test = 321.0 (*p*-value < 0.01). The null hypothesis is rejected at the significance of 1%. It indicates that the neighboring pixels' urban surface features ($WX$) is significantly not 0 in the model, and the SDM model should not be simplified to the SLM model. In other words, the model specification with the neighboring pixels' urban surface features ($WX$) is appropriate, which is closer to the real generation process of NTL.

In summary, when the model specification is the SDM model, it is closest to the real generation process of NTL.

### 4.2.4. Comparison between Different Models

The comparison between the SDM model and the other five models was made clearer by certain statistics. The Moran test was performed on the regression residuals of all six models (Table 4). It was obvious that the regression residuals of the OLS model and the SLX model featured a fairly strong spatial autocorrelation, at a significance level of 1%. The regression residuals of the SDM model, SDEM model, SLM model and SEM model featured a weak spatial autocorrelation, which was close to spatial random distribution. This indicated that the problem of independent residuals was solved by the model specification with the neighboring pixel's NTL ($WY$) and the neighboring pixel's urban surface features ($WX$). In other words, the neighboring effect or PIBE of the NTL intensity was completed explained.

**Table 4.** Comparison of goodness of fit between different models.

| Model | Moran's $I$ | Pearson Correlation (Squared) | $R^2$ | Log Likelihood | AIC |
|---|---|---|---|---|---|
| OLS | 0.685 ** | 0.674 (0.454 #) | 0.454 | −35,838.7 | 71,701.4 |
| SEM | −0.013 ** | 0.618 (0.382 #) | 0.914 | −11,889.8 | 23,803.7 |
| SLM | −0.002 * | 0.769 (0.592 #) | 0.915 | −10,844.2 | 21,714.5 |
| SLX | 0.732 ** | 0.751 (0.564 #) | 0.563 | −32,236.2 | 64,518.3 |
| SDEM | −0.016 ** | 0.736 (0.542 #) | 0.915 | −11,139.9 | 22,326.1 |
| SDM | 0.014 ** | 0.774 (0.600 #) | 0.915 | −10,718.9 | 21,485.8 |

* represents *p*-value < 0.05; ** represents *p*-value < 0.01; # represents the square of Pearson correlation coefficient.

Due to the fact that the pseudo-$R^2$ of the spatial autoregressive model was different from the $R^2$ of the OLS model, the $R^2$ of different models could not be compared. We applied three statistics to compare the models: log likelihood, Akaike information criterion (AIC) and Pearson correlation coefficient. The larger the log likelihood, the smaller the AIC, and the better the model. Pearson correlation coefficient and its square are the statistics between the estimated value and the observed value of the NTL intensity. According to Table 4, the SDM model is the optimal model.

### 4.3. SDM Fitting

The SDM model was estimated by Luojia1-01 NTL intensity and urban surface features (a total of 13 variables from Natural surface, POIs and roads) at pixel level. The NTL inten-

sity of Luojia1-01 was effectively explained (pseudo-$R^2$ = 0.915; Pearson correlation = 0.774), and its PIBE was effectively explained by the urban surface features (Moran's I = 0.014). Figure 6 illustrates the comparison between the NTL intensity observed by Luojia1-01 (Figure 6a) and the NTL intensity simulated by the SDM model (Figure 6b). Furthermore, it illustrates the local Moran's I between the regression residual of the OLS model (Figure 6c) and the SDM model (Figure 6d). The results of the local Moran's I indicate that there was no significant spatial autocorrelation in the residual of the SDM model, and the PIBE between different pixels was completely explained. The independent variables of government and social organization and public garden were not significant in the model and were eliminated (Table S2). The reason is that the sample size of POIs in the corresponding category was too small. The Pearson correlation coefficient of the SDM model was 0.774. This indicated that it was feasible to measure the NTL intensity and PIBE by POIs, OSM road network, water body and vegetation.

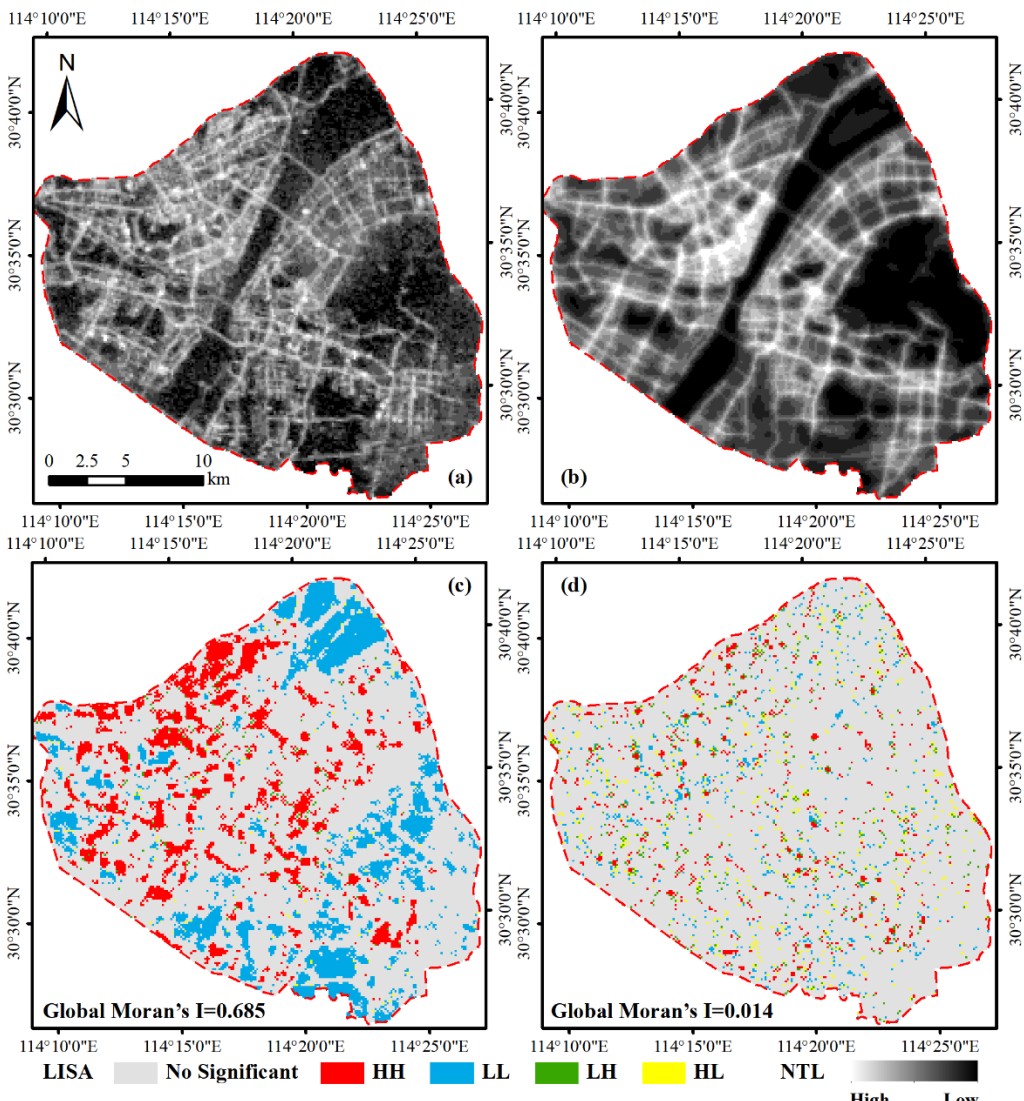

**Figure 6.** (**a**) NTL intensity observed by Luojia 1-01 image; (**b**) NTL intensity simulated by SDM model; (**c**) local Moran's I of OLS model regression residual; (**d**) local Moran's I of SDM model regression residual. HH represents the aggregation of high values and high values. LL represents the aggregation of low values and low values. LH represents low values surrounded by high values. HL represents high values surrounded by low values. No Significant represents the residuals' approximately random distribution.

### 4.4. Spatial Partitioning of Feature Contributions

The estimation results of the SDM model are shown in Table S2, but the total/direct/indirect effect (Table 5) were generally used instead of the regression coefficient (Table S2) to measure the contributions. Table 5 displays the total effect, direct effect and indirect effect of different independent variables in the SDM model. The total effect of road1 was 0.508: for any center pixel, 1 standard deviation (SD) increase in the road1 contributed a 0.508 SD increase to the NTL intensity. A total of 12% of this contribution derived from the direct effect and contributed a 0.061 SD increase to the center pixel's NTL intensity. A total of 88% of this contribution derived from the indirect effect and contributed a 0.447 SD increase to the PIBE (including low-order neighbor and high-order neighbor). The total effect of vegetation was −0.958: when any center pixel changed from non-vegetation to vegetation, the NTL intensity would decrease 0.958 SD. A total of 8% of this contribution derived from the direct effect and contributed a 0.077 SD decrease to the center pixel's NTL. 92% of this contribution comes from the indirect effect and contributes 0.881 SD decrease to the PIBE. The contributions of different urban surface features can be compared, because the model is set in standardized mode. This comparison does not include vegetation and water because they were used as dummy variables. According to Table 5, the contribution of roads decreased with the reduction in the road grade (road5 was the lowest). The total, direct and indirect contributions of vegetation and water were significantly negative, but the inhibitory effect of water on PIBE was weaker than that of vegetation (absolute value of indirect effect: 0.800 < 0.881).

**Table 5.** Contributions of different urban surface features to Luojia 1-01 NTL.

| Urban Surface Features | Total Effect | Direct Effect | Indirect Effect |
|---|---|---|---|
| Residential | −0.137 ** | −0.011 ** (7.8%) # | −0.127 * (92.2%) # |
| Commercial | 0.098 *** | 0.023 *** (23.4%) | 0.075 ** (76.6%) |
| Industrial | −0.073 * | −0.005 (7.0%) | −0.068 * (93.0%) |
| Transportation facilities | 0.299 *** | 0.025 *** (8.5%) | 0.273 *** (91.5%) |
| Road1 | 0.508 *** | 0.061 *** (12.0%) | 0.447 *** (88.0%) |
| Road2 | 0.662 *** | 0.059 *** (9.0%) | 0.603 *** (91.0%) |
| Road3 | 0.567 *** | 0.057 *** (10.1%) | 0.510 *** (89.9%) |
| Road4 | 0.285 *** | 0.035 *** (12.4%) | 0.250 *** (87.6%) |
| Road5 | 0.184 *** | 0.024 *** (13.2%) | 0.160 *** (86.8%) |
| Water | −0.879 *** | −0.078 *** (8.9%) | −0.800 *** (91.1%) |
| Vegetation | −0.958 *** | −0.077 *** (8.0%) | −0.881 *** (92.0%) |

(1) * represents *p*-value < 0.1; ** represents *p*-value < 0.05; *** represents *p*-value < 0.01. (2) #: for the direct effect, the value of direct effect is outside the bracket, and the percentage of direct effect in total effect is inside the bracket. The same goes for indirect effect. (3) The effects from an observation and the effects to an observation are numerically equal, because the spatial weight matrix in this study is a symmetric matrix.

The impact of feature contributions on low-order neighbors was higher than on high-order neighbors. When the total effect is partitioned by the order of $W$, more information is available [37]. The higher-order spatial weight matrix refers to the exponential power of $W$, such as $W^2$, $W^3$... (in Figure 7). We can partition the total effect in space to illustrate its characteristics when the effect moving to higher-order neighbors (Table 6). This is worthwhile when the spatial extent and the attenuation pattern of the PIBE are the purpose.

When the feature contribution is partitioned by the order of $W$, more information is available. Table 6 displays the marginal direct effect and marginal indirect effect of road2. Some additional information can be summarized, which is consistent in different urban surface features. (i) The direct effect (0.0594) was greater than the marginal indirect effect of each neighbor, but far less than the indirect effect (0.6030, 5–10 times of the direct effect). This indicated that the contribution of the center pixel's urban surface feature to the center pixel's NTL intensity was greater than that to PIBE of each neighboring pixel, but far less than the aggregate PIBE of all neighboring pixels. (ii) The direct effect soon disappeared with the movement to the higher-order neighbors, while the attenuation speed of the indirect effect was much slower. The values of the marginal direct effect and marginal

indirect effect had no practical significance for the higher-order neighbors. (iii) When the cumulative proportion was set to 90% as the threshold of the effect, the road2's attenuation distance of the indirect effect was about $W^{24}$, and the direct effect was about $W^8$. According to this standard, Figure 8 illustrates the attenuation distance of the direct effect and indirect effect of different urban surface features. The attenuation distance of the indirect effect represents the attenuation distance of PIBE. Different urban surface features reached 90% cumulative PIBE around 3 km ($W^{24}$, 3120 m, convert by pixel size 130 m × 130 m).

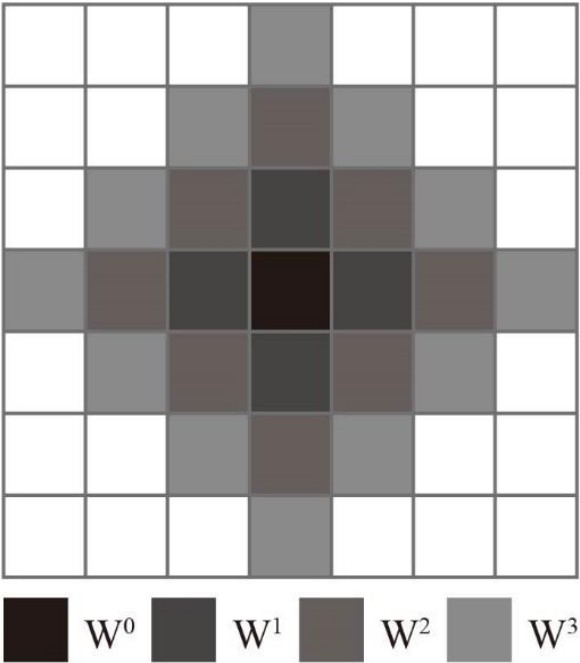

**Figure 7.** Schematic diagram of higher-order rook spatial weight matrix. $W^0$ represents center pixel; $W^1$ represents first-order neighboring pixels; $W^2$ represents the neighbor of $W^1$.

**Table 6.** Spatial partitioning of road2's contributions.

| Order | Marginal Direct Effect | Cumulative Percent | Marginal Indirect Effect | Cumulative Percent |
|---|---|---|---|---|
| $W^0$ | 0.0278 | 46.8% | 0.0355 | 5.9% |
| $W^1$ | 0.0081 | 60.5% | 0.0490 | 14.0% |
| $W^2$ | 0.0057 | 70.0% | 0.0460 | 21.7% |
| $W^3$ | 0.0038 | 76.4% | 0.0427 | 28.8% |
| $W^4$ | 0.0026 | 80.8% | 0.0397 | 35.3% |
| $W^5$ | 0.0021 | 84.4% | 0.0359 | 41.3% |
| $W^6$ | 0.0016 | 87.0% | 0.0328 | 46.7% |
| $W^7$ | 0.0013 | 89.2% | 0.0297 | 51.7% |
| $W^8$ | 0.0010 | 90.9% | 0.0274 | 56.2% |
| $W^9 \dots W^{23}$ | … … | … … | … … | … … |
| $W^{24}$ | … … | … … | 0.0060 | 90.8% |
| Cumulative | 0.0594 | 100.00% | 0.6030 | 100.00% |

Note: The spatial partitioned contribution is explained from the perspective of "the impact from an observation". For any center pixel, a 1 SD increase in the road2 contributes a 0.0594 SD increase to the center pixel's NTL intensity. The marginal direct effect of higher-order neighbors represents the feedback effect (such as pixel $i \rightarrow j \rightarrow i$). Concurrently, a 1 SD increase in the road2 contributes a 0.6030 SD increase to PIBE. Among these, a 0.0490 SD increase derives from the first-order neighbors' NTL intensity; a 0.0460 SD increase derives from the second-order neighbors' NTL intensity; and so on. The marginal indirect effect eventually spreads to the whole study area. After accumulation, it is equal to the indirect effect [37].

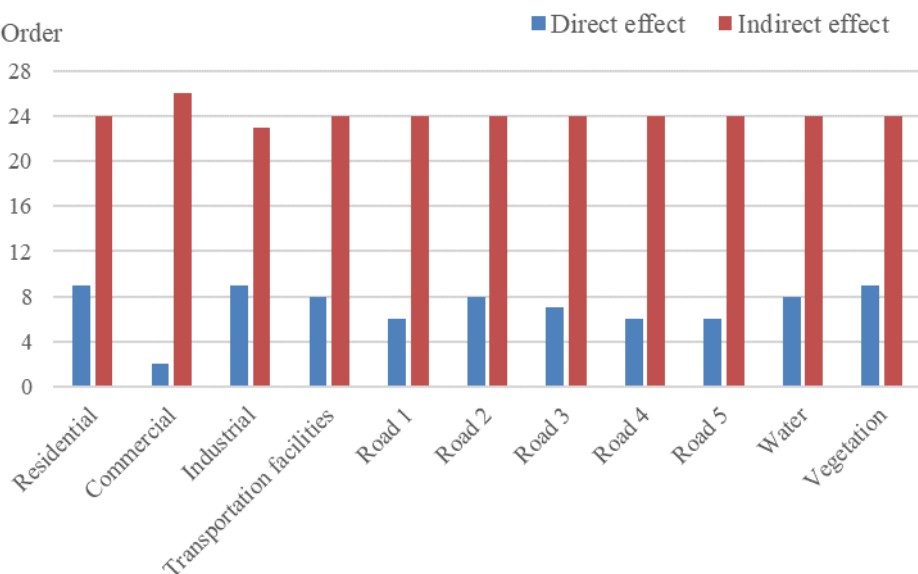

**Figure 8.** The attenuation distance of the direct effect and the indirect effect.

## 5. Discussion

### 5.1. Urban Surface Feature Contributions to Luojia 1-01 NTL Intensity

The contributions of different urban surface features to Luojia1-01 NTL intensity and PIBE are significantly different. Roads and transportation facilities provide the major contribution to NTL intensity and PIBE. Commercial is slightly lower. The attenuation patterns of PIBE with different urban surface features are roughly the same. The contribution of the center pixel's urban surface feature to the center pixel's NTL intensity is greater than to the PIBE of each neighboring pixel, but far less than the aggregate PIBE of all the neighboring pixels. The attenuation distance of the PIBE with different urban surface features is roughly the same, about 3 km. The PIBE of the high-order neighboring pixels has no practical significance.

Roads provide the highest total contribution (0.184~0.508) and direct contribution (0.024~0.061) to NTL intensity, and their PIBE (indirect contribution = 0.160 ~ 0.603) show a strong positive spillover effect. They are the major source of urban NTL. The direct contribution of roads is significantly positive, and is much higher than that of other urban surface features. The contributions of different-grade roads to NTL intensity and PIBE are clearly different. The contributions of roads to NTL intensity generally increase with the increase in road grade (road5 is the lowest). However, the PIBE of road1 is slightly weaker. When the roads are clustered, the road not only contributes greatly to the NTL in the current area, but also shows a strong positive spillover effect, that is, clear lights can also be observed in the surrounding area. Due to the strong interaction between the road and the outside, most of the interaction carriers are vehicles [5], and the contribution of the road to PIBE is further increased.

Transportation facilities provide a positive total contribution (0.299) to NTL intensity, and their PIBE (indirect contribution = 0.273) shows a strong positive spillover effect. Their indirect contribution is greater than that of commercial (0.273 > 0.075). The transportation facilities are artificial facilities, such as parking lots, bus stops and subway stations. Their direct and indirect contribution to NTL intensity is equivalent to that of road4. Due to the open-air characteristics of artificial facilities such as parking lots and bus stops, the light propagation is less restrained. This makes it possible to expand the light extent that can be detected by the Luojia1-01.

Residential and industrial provide a negative total contribution (−0.137, −0.073) to NTL intensity, which belongs to the shadow in the urban interior. When these two kinds of urban surface features cluster together, they not only make no obvious contribution to the

NTL intensity of the center pixel, but also reduce the NTL intensity of all the neighbors due to the negative indirect effect. As the overpass time is 22:41, it is close to the rest time of the residential, the intensity of human activities is limited and the outward spillover capacity is limited. The population aggregation of residential may not produce more human activities, which makes its PIBE limited. The contribution of industrial to NTL intensity is not obvious in this study area, which is consistent with previous studies [9,41]. The results of our model are consistent with the human activities of residential and industrial in the urban interior.

Commercial provides a positive total contribution (0.098) to NTL intensity, and its PIBE (0.075) is lower than that of roads. Its contribution to NTL intensity is the highest except for roads and transportation facilities. Commercial includes catering, shopping, companies, hotels and office buildings. Its direct contribution to NTL intensity is significant, but its indirect contribution is less so. Since the overpass time is 22:41 and the curfew time stipulated by the urban management department is 22:00, most of the commercial lights on the facade of buildings are turned off [42]. Due to the obstruction of light propagation by roofs and surrounding buildings [43], the light that can be detected by Luojia1-01 is reduced, which further reduces the contribution of commercial to NTL intensity. In POI data, small commercial buildings are equivalent to large commercial buildings; the proportion of the former is much higher than that of the latter. However, the contributions of small commercial buildings to the NTL intensity and the PIBE are weak. Therefore, the direct contribution of commercial to NTL intensity is roughly the same as that of road5, and the indirect contribution is higher than that of residential and industrial.

Vegetation and water provide a negative total contribution (−0.958, −0.879) to NTL intensity, and their PIBE (−0.881, −0.800) show a strong negative spillover effect. They intensely inhibit NTL intensity and PIBE, which is the major shadow area in the urban interior. However, the absolute total contribution of water is less than that of vegetation (0.879 < 0.958), that is, the inhibitory effect of water on NTL intensity is weaker than that of vegetation. When the neighbor clusters vegetation and water, human activities and light in the central pixel are inhibited, which makes the PIBE of vegetation and water significantly negative. The absolute value of the indirect contribution of water to NTL intensity is smaller than that of vegetation (0.800 < 0.881). Due to the fact that water reflects considerable light [25], the inhibition effect of water on PIBE is weaker than that of vegetation. The contribution of vegetation and water to NTL intensity is consistent with human activities in its corresponding area. Our result indicates that vegetation and water can greatly inhibit light pollution in the urban interior.

*5.2. Neighboring Pixel's Effect*

It is worth noting that the $WY$ in our model is numerically equal to the average of the neighboring pixels' NTL intensity, but it should not be regarded as an independent variable of $Y$ [37]. $WY$ is the model parameter to solve the spillover effect. Its coefficient ($\rho$) is used as a global multiplier for each independent variable by partial derivative, which is reflected in the indirect effect. The difference between the SDM model and the SLX model is $WY$, which makes the SLX model ignore the transfer effects between neighboring pixels [40].

The $WX$ is regarded as the independent variable of $Y$ in this study. In the spatial data, the neighboring pixels' independent variable ($WX$) usually had a directly perceptible impact on the center pixel (including positive and negative). For instance, the characteristics of the urban surface illuminants (the brightness intensity [21,23] of the illuminants affected the blooming effect) and the background environment of the urban surface illuminants (the vegetation had an inhibitory effect on the NTL [24], and the blooming effect was more significant when it was close to water and snow [25]). Previous studies have supported the rationality of $WX$ from theoretical and practical perspectives. This article further proves the statistical significance of $WX$ by hypothesis test.

The definition of "bloom effect" in previous studies may not standardize. It usually describes a phenomenon that actual lit areas to be enlarged in NTL image. This means that the mechanism of the "bloom effect" may be different in various platforms, sensors or

spatial resolutions. The concept of PIBE proposed in this paper focuses on the pixel interaction in Luojia1-01 remote sensing NTL images. This paper places emphasizes the NTL variation of the central pixel coursed by the variation of the neighboring pixels in statistics. For various data collected in different ways (such as DMSP/OLS, NPP/VIIRS, Luojia1-01, unmanned aerial vehicle, digital single lens reflex camera), it will be a very meaningful work to explore the difference, physical mechanism and laws of this phenomenon. We will try to conduct more in-depth research in future studies.

*5.3. Limitation*

A limitation of spatial autoregressive models is the preset spatial weight matrix. As an important parameter with which to characterize the spatial relationship between units in spatial autoregressive models, there is no accurate method to estimate the spatial weight in the current research. In this study, several common spatial weight matrices (i.e., Queen, Rook and inverse distance weight) was included in the model, which screened by the loglikelihood [44]. It is concluded that the optimal spatial weight matrix is the Rook spatial weight matrix. In addition, when the coverage distance of different spatial weight matrices is roughly the same, there is little change in the PIBE and its attenuation pattern. We will try to explore more weight matrices in future research.

## 6. Conclusions

Previous studies continuously emphasized the importance of NTL interaction between different units in theory [4,21,26]. This interaction is clearer at pixel level [21,26]. We propose a method through which to explain and explore the spatial distribution of NTL by considering the effect of neighboring pixels. This article tested the statistical significance of NTL interactions between different pixels, and the theory of this interaction is consistent with the hypothesis test. Our method can analyze the differential contributions of different urban surfaces to NTL intensity, and effectively partition the contributions of different urban surface features to the NTL intensity and the PIBE (pseudo-$R^2$ = 0.915; person correlation = 0.774, Moran's I = 0.014). We can receive more information about the relationships between pixels through the direct effect and the indirect effect. That is, we can determine how much of the central pixel's NTL is "spilled" from its neighboring pixels and how much is caused by the central pixel's urban surface. Furthermore, in this paper, the response between urban surface features and Luojia 1-01 NTL intensity was explored and promoted to the pixel level. Compared with previous studies, this enables us to reveal the human activities that can explain the NTL variations on a finer scale. The method proposed in this study is expected to provide a reference for explaining the composition and blooming effect of NTL, as well as the application of NTL data in the urban interior.

The in-depth analysis of urban interior space by NTL remote sensing data is a gordian knot at present and will definitely be a significant topic in the future. For future applications, combining NTL data with multisource data (such as POI, OSM, taxi track data, mobile signal data and other open big data) will be a major direction for exploring urban interior structure. Measuring the contributions of different urban land use to NTL intensity and the blooming effect is of great value to the extraction of build-up area and the inversion of grid population data inversion using NTL data.

**Supplementary Materials:** The following are available online at https://www.mdpi.com/article/10.3390/rs13234838/s1. Table S1: Spatial Durbin model and the equations of its total/direct/indirect effect; Table S2: Estimation results of OLS and SDM models.

**Author Contributions:** J.W. and X.L. conceived and designed the experiments; X.Y. and Z.Z. provided guidance on the methodology; J.W. and X.L. performed the experiments and wrote the paper; and all authors edited the paper. All authors have read and agreed to the published version of the manuscript.

**Funding:** This work was supported by the National Key R&D Program of China under Grant 2019YFE0126800, the National Natural Science Foundation of China under Grant 41771386, the Fun-

damental Research Funds for the Central Universities under Grant 2042021kf1053, the open project of key laboratory of geological processes and mineral resources in Northern Tibet Plateau of Qinghai Province under Grant 2019-KZ-01 and the special project for innovation platform construction of science and technology department of Qinghai Province under Grant 2019-ZJ-T04.

**Acknowledgments:** We give thanks to the editors and anonymous reviewers for their valuable comments to improve our manuscript.

**Conflicts of Interest:** The authors declare no conflict of interest.

## Appendix A

**Table A1.** Types of original and aggregated POIs.

| Aggregated Type | Initial Type of Amap |
|---|---|
| Residential | Commercial House * (except Industrial Park, Building **) |
| Government and Social organization | Medical Service<br>Governmental Organization and Social Group<br>Science/Culture and Education Service<br>(except Media Organization, Training Institution, Driving School)<br>Public Facility<br>Daily Life Service (including Job Center, Funeral Facilities)<br>Sports and Recreation (including Sports Stadium) |
| Commercial | Food and Beverages<br>Shopping<br>Daily Life Service (except Job Center, Funeral Facilities)<br>Auto Service<br>Sports and Recreation (except Sports Stadium)<br>Accommodation Service<br>Finance and Insurance Service<br>Enterprises<br>(except Factory, Company-Chemical and Metallurgy, Company-Machinery and Electronics)<br>Medical Service (including Clinic, Veterinary Hospital)<br>Science/Culture and Education Service<br>(including Media Organization, Training Institution, Driving School)<br>Commercial House (including Building) |
| Industrial | Commercial House (including Industrial Park)<br>Enterprises<br>(including Factory, Company – Chemical and Metallurgy, Company – Machinery and Electronics) |
| Transportation facilities | Transportation Service |
| Public garden | Tourist Attraction |

* outside the bracket is the big category of Amap; ** inside the bracket is the mid category of Amap. POIs that do not belong to the above types are deleted.

**Table A2.** Types of original and aggregated roads.

| Aggregated Type | Initial Type of OSM |
|---|---|
| Road1 | motorway, trunk, motor way_link, trunk_link |
| Road2 | primary way, primary way_link |
| Road3 | secondary way, secondary way_link |
| Road4 | tertiary way, tertiary way_link |
| Road5 | residential, and others (such as cycleway, footway, living_street, path, pedestrian) |

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
