# Peer review of "Analyzing Pixel-Level Relationships between Luojia 1-01 Nighttime Light and Urban Surface Features by Separating the Pixel Blooming Effect"

_remotesensing, doi:10.3390/rs13234838_

Round 1
Reviewer 1 Report
Overall comments:
This paper presents an interesting topic by separating direct and indirect effects (PIBE; pixel blooming effect) on pixel-level nighttime light intensity observed by Luojia1-01. The authors also analyze the relationships between these two effects and urban surface features. I agree that this paper is useful for the monitoring of urban socio-economic activities. Some comments are provided below to further improve this paper:
Although this paper stresses separating direct and in-direct/blooming effects on NTL, it does not mention this point in the title. The title is too general and makes this paper look similar to other published articles despite that it has some new findings. I suggest refining the title of this article to highlight the main points of the text.
Specific comments:
Lines 41, 128, 133: Please show the full names of NTL/PIBE/SDM for the first time use in the main body of the article although it has been mentioned in the abstract.
Line 59: Please add a space before the in-text references.
Lines 120-125: I suggest simplifying sentences in this paragraph and merging them with the above paragraph because both paragraphs are talking about challenges relating to the blooming effect.
Lines 468-475: I suggest moving this paragraph to the method section.
Table 5: I suggest adding new columns to present the percentage of direct/indirect effect contributions.
Line 688: The journal name for this article seems not correct.
Lines 701 & 709: I think it’s okay to just use ‘Remote sensing’ here. Please keep consistent with this journal name across the reference list.
Reviewer 2 Report
I found the manuscript very interesting as I also work with nighttime lights data. The work is described with scientific rigor and sound statistical techniques. I take issue with the term "blooming" as applied to VIIRS nighttime lights. While true of DMSP nighttimelights, the VIIRS Day/Night band does not saturate. I think the authors may be referring to the characteristics of a scanning radiometer instrument to capture some light from neighboring pixels, but this isn't what I would call "blooming". It is true that this effect is present for water and snow however, as mentioned in the paper. It would be preferable for the authors to be more specific about this effect, and the difference between VIIRS and DMSP. Otherwise, I think the paper makes an important contribution to the nighttime lights research literature.
Reviewer 3 Report
The authors propose an approach to explore the spatial distribution of NTL and assess the pixel blooming effect by superimposing panchromatic Luojia 1-01 night-time light data, as well as data of 13 types of land-uses. I highly appreciate the preciseness with which the authors explained their approach. I recommend publishing the article after addressing several minor issues:
1) I lack the explanation why "For NTL images with coarser spatial resolution, the blooming effect is more obvious" (line 75)
2) As far as I know, both DMSP and VIIRS report NTL in the diapason of 500-900 nm. In Table 1, you report other numbers. Please check.
3) In the text, I detected some repetitions (see lines 326-328 and 372-374).
4) You write that "The independent variables of government & social organization and public garden are not significant in the model and have been eliminated in the model" (lines 455-456). I would like to see the models themselves. And I did not find an explanation of where the coefficients in Table 5 come from.
Author Response
Please see the attachment.

This manuscript is a resubmission of an earlier submission. The following is a list of the peer review reports and author responses from that submission.